# Genomoviruses in Liver Samples of *Molossus molossus* Bats

**DOI:** 10.3390/microorganisms12040688

**Published:** 2024-03-29

**Authors:** Roseane da Silva Couto, Wandercleyson Uchôa Abreu, Luís Reginaldo Ribeiro Rodrigues, Luis Fernando Marinho, Vanessa dos Santos Morais, Fabiola Villanova, Ramendra Pati Pandey, Xutao Deng, Eric Delwart, Antonio Charlys da Costa, Elcio Leal

**Affiliations:** 1Laboratório de Diversidade Viral, Instituto de Ciências Biológicas, Universidade Federal do Pará, Belem 66075-000, PA, Brazil; couto.roseane@gmail.com (R.d.S.C.); fvillanova@gmail.com (F.V.); 2Programa de Pos-Graduação REDE Bionorte, Polo Pará, Universidade Federal do Oeste do Pará, Santarém 68040-255, PA, Brazil; uchoa_vet@yahoo.com.br; 3Laboratory of Genetics & Biodiversity, Institute of Educational Sciences, Universidade Federal do Oeste do Pará, Santarém 68040-255, PA, Brazil; luisreginaldo.ufpa@hotmail.com; 4University of Amazonia, Santarém 68040-255, PA, Brazil; fm8885785@gmail.com; 5Laboratory of Virology (LIM 52), Instituto de Medicina Tropical, Universidade de São Paulo, São Paulo 05403-000, SP, Brazil; va.morais@usp.br (V.d.S.M.); charlysbr@yahoo.com.br (A.C.d.C.); 6School of Health Sciences & Technology, UPES University, Dehradun 248007, Uttarakhand, India; ramendra.pandey@ddn.upes.ac.in; 7Vitalant Research Institute, San Francisco, CA 94143, USA; xutaodeng@gmail.com; 8Department Laboratory Medicine, University of California San Francisco, San Francisco, CA 94143, USA; edelwart@vitalant.org

**Keywords:** velvety free-tailed bat, CRESS-DNA, *Genomoviridae*, *Gemykibivirus*

## Abstract

CRESS-DNA encompasses a broad spectrum of viruses documented across diverse organisms such as animals, plants, diatoms, fungi, and marine invertebrates. Despite this prevalence, the full extent of these viruses’ impact on the environment and their respective hosts remains incompletely understood. Furthermore, an increasing number of viruses within this category lack detailed characterization. This investigation focuses on unveiling and characterizing viruses affiliated with the *Genomoviridae* family identified in liver samples from the bat *Molossus molossus*. Leveraging viral metagenomics, we identified seven sequences (MmGmV-PA) featuring a circular DNA genome housing two ORFs encoding replication-associated protein (Rep) and capsid protein (Cap). Predictions based on conserved domains typical of the *Genomoviridae* family were established. Phylogenetic analysis revealed the segregation of these sequences into two clades aligning with the genera *Gemycirculavirus* (MmGmV-06-PA and MmGmV-07-PA) and *Gemykibivirus* (MmGmV-01-PA, MmGmV-02-PA, MmGmV-03-PA, MmGmV-05-PA, and MmGmV-09-PA). At the species level, pairwise comparisons based on complete nucleotide sequences indicated the potential existence of three novel species. In summary, our study significantly contributes to an enhanced understanding of the diversity of *Genomoviridae* within bat samples, shedding light on previously undiscovered viral entities and their potential ecological implications.

## 1. Introduction

Bats are large natural reservoirs of zoonotic viruses that can cause various viral diseases in mammals as well as in humans [1,2]. Over 20 years, several zoonotic outbreaks with major impacts emerged that led to the deaths of thousands of humans associated with the virome in bats, such as SARS-CoV-2 [3], Nipah [4], Hendra [5], and Herpesvirus [6]. A wide variety of new virus species in bats have been identified and characterized using metagenomics. This technology has helped in several areas, such as precision medicine to diagnose infectious diseases and maintain public health alerts [7,8].

The swift progress of next-generation sequencing (NGS) technology coupled with metagenomics has proven adept at unveiling a spectrum of novel circular ssDNA viruses encoding the Rep protein (Circular Rep-encoding Single-Stranded DNA) or RNA viruses [9]. These advancements extend across various specimen categories, encompassing fecal samples [10], heterogeneous mixtures [11], human subjects [12,13,14], bat populations [15], turtles [16], avian species [17], insects [18,19], plants [20], parasites [21,22], and environmental matrices [23]. This technological synergy facilitates the isolation and comprehensive characterization of viral diversity within diverse ecosystems.

In this framework, the International Committee on Taxonomy of Viruses (ICTV), (accessible at https://ictv.global/taxonomy, accessed on 5 June 2023) assumes a pivotal role. It not only acknowledges but also establishes the standardization of viral species’ nomenclature, employing the binomial system (genus and epithet) for all viruses [24]. Classifying a species within the *Genomaviridae* family into one of the existing 10 genera mandates a phylogenetic analysis grounded in the Rep protein. To accomplish this, the ICTV imposes a criterion wherein a <78% identity across the entire genome denotes a novel species, and similar thresholds apply to Rep for genre assignment [25]. This systematic approach ensures precision in delineating viral taxonomy, offering a comprehensive framework for understanding and categorizing the evolving landscape of *Genomaviridae*.

CRESS-DNA viruses infect eukaryotes and are classified in the phylum Cressdnaviricota, in two classes: *Arfiviricetese* and *Repensiviricetes*. The *Arfiviricetese* class contains 12 families (*Amesuviridae*, *Bacilladnaviridae*, *Circoviridae*, *Metaxyviridae*, *Nanoviridae*, *Naryaviridae*, *Nenyaviridae*, *Redondoviridae*, *Smacoviridae*, *Vilyaviridae*). Repensiviricetes contains only two families: *Geminiviridae* and *Genomoviridae* [26,27].

The *Genomoviridae* family is composed of 10 genera and 237 species distributed among *Gemycircularvirus* (126 species), *Gemyduguivirus* (12 species), *Gemygorvirus* (eight species), *Gemykibivirus* (50 species), *Gemykolovrus* (16 species), *Gemykrogvirus* (13 species), *Gemykroznavirus* (seven species), *Gemytondvirus* (one species), *Gemytripvirus* (one species), and *Gemyvongvirus* (three species) [25,28]. In general, genomes are small and circular (~1.8–2.4 kb), monopartite, single-stranded ssDNA, and non-enveloped. They contain two open reading frames (ORFs): one encoding the replication-associated protein (Rep) on the complementary strand, and the other encoding the capsid protein (Cap) on the viral strand. Rep is a multifunctional protein containing an endonuclease and a helicase domain. The N-terminal endonuclease domain has three stretches of highly conserved sequences identified as motifs [29]: Motif I (uuTYxQ), Motif II (xHxHx) [30,31], and Motif III (YxxK) [31,32], with the letter (u) indicating bulky hydrophobic residues such as I, L, V, M, F, Y, and W, and the letter (x) representing any residue [33]. The GRS motif is located between motifs II and III [31]. Rep’s helicase activity is measured by conserved motifs such as Walker A (GxxxxGKT), Walker B (uuDDu), and the C-terminal motif (UxxN) [33,34,35].

Initially, the *Genomaviridae* family housed a solitary genus, *Gemycircularvirus,* and a lone species, *Sclerotinia gemycirculavirus1* (SsHADV-1) [36]. Members of this viral family have been extracted from a wide array of hosts, spanning fungi, animals, plants, diatoms, and insects, with recent characterizations extending to various species of marine invertebrates, particularly crustaceans. Viral metagenomic studies have indicated that CRESS-DNA viruses, such as Geomovirus, are associated with different hosts, such as protozoa of the genera Giardia [21] and Entamoeba [22].

Despite these discoveries, a substantial number of CRESS-DNA viruses remain unclassified, even in protozoa *Giardia*, belonging to one family (*Vilyaviridae*) and *Entamoeba* in two virus families (*Naryaviridae* and *Nenyaviridae* [21,22]. It is noteworthy that CRESS viruses do not classify; 71% resemble chimeric Reps (replication-associated proteins encoded by single-stranded DNA viruses (CRESS), which encode proteins associated with circular replication (Rep), since there is the constant exchange of gene fragments that encode the nuclease and helicase domains. Thus, unclassified CRESS viruses present intense chimerism in their Reps [27] and their impact on the environment and their respective hosts remains poorly understood [26].

The identification of novel single-stranded DNA (ssDNA) viruses akin to SsHADV-1, observed across diverse environmental and animal samples, has brought to light a previously unexplored realm of viral biodiversity [13,28,37]. The *Gemycircularvirus* genus boasts the highest count of viral species within the *Genomaviridae* family, closely followed by the *Gemykibivirus* [28]. In this study, we present the comprehensive characterization of seven complete sequences belonging to the *Genomoviridae* family, stratified into two genera (*Gemykibivirus* and *Gemycircularvirus)*, along with the identification of three novel species. These findings were derived from bats of the *Molossus molossus* species in the Northern region of Brazil.

## 2. Materials and Methods

### 2.1. Sample Collection

In Caranazal, situated at latitude 2°26′10′′ S and longitude 54°43′49′′ W in the Santarém region of Pará state within the Lower Amazon Mesoregion, we captured 47 bats. These bats were identified as *Molossus molossus* from the *Molossidae* family based on external characteristics [28]. The process involved euthanizing individual bats for sample collection, employing xylazine hydrochloride (1 mg/kg) and ketamine hydrochloride (1–2 mg/kg) via intramuscular injection for anesthesia, followed by intracardiac phenobarbital (40 mg/kg) administration once unconsciousness was achieved.

Approval for this research was obtained from the Animal Use Ethics Committee of the Federal University of Western Pará (CEUA/UFOPA) under number 0220220128, and capturing Chiroptera was authorized by the Biodiversity Information and Authorization System. The necropsy procedures were conducted at the Animal Morphology Laboratory of the Federal University of Western Pará, adhering to institutional biosafety norms.

### 2.2. Processing of Samples

Due to a shortage of reagents, we used pieces of all 47 samples in a single pool. The processing of samples involved a systematic approach to isolate viral particles from liver tissue. Genomoviruses have tropism for liver cells, meaning they are found in greater quantities in liver tissues than in other tissues. When we use tissue in which the virus has lower tropism, we will find a lower concentration of viruses. Therefore, in this tissue, the amount of other microorganisms can interfere with data analysis. The tissue extraction process began with specimen maceration, followed by dilution in 500 µL of Hanks Buffered Saline Solution (HBSS). Subsequent homogenization using a vortex mixer occurred in 2 mL tubes containing lysis matrix C (MP Biomedicals, Santa Ana, CA, USA). After filtration through 0.45 µM filters (Merck Millipore, Billerica, MA, USA) to eliminate eukaryotic and bacterial cell particles, clarified filtrates underwent centrifugation at 32,000 rpm for 1 h. This facilitated the sedimentation of viral particles, which were then carefully resuspended in 250 µL of PBS for nuclease enzyme treatment.

To enhance the purity of the viral isolates, the filtrates underwent treatment with DNase and RNase A at 37 °C for 30 min, followed by Phi29 (Φ29) polymerase enzyme treatment for DNA circular amplification. The entire extraction process, characterized by precise steps and strategic enzymatic treatments, ensured the isolation of high-quality viral particles from liver tissue.

### 2.3. Nucleic Acid Extraction (DNA/RNA)

For sample preparation, we utilized the QIAamp Viral RNA Mini Kit (GmbH, QIAGEN Strasse 1, 40724 Hilden, Germany) according to the instructions of the manufacturer.

### 2.4. Preparation of Libraries for the Illumina Platform

Library preparation for the Illumina platform was performed using the Nextera XT DNA Sample Preparation Kit (Illumina Inc., San Diego, CA, USA), and sequencing was carried out on the Illumina NovaSeq-6000 platform, producing 250 bp paired reads.

### 2.5. Bioinformatics Analysis

For data analysis, raw reads underwent pre-processing to remove terminal-matched sequence records, exclude low-quality sequences, and eliminate adapter and primer sequences. The subsequent bioinformatic analysis confirmed the specificity of the analysis, revealing no reads associated with human, plant, fungal, or bacterial sequences. Contigs were compared against the GenBank genetic sequence database using BLASTx and BLASTn.

The predicted gene sequences obtained from BLAST searches were meticulously chosen based on the most favorable outcomes. To confirm and further classify these sequences, reads and/or contigs were aligned against a viral protein database using DIAMOND software version 2.1.8 (double index alignment of next-generation sequencing data). DIAMOND compares translated DNA sequences in pairs of proteins and offers various output formats, including tabular, paired BLAST, and XML. This process aids in refining the taxonomic classification of viruses. It is worth noting that DIAMOND is an open-source algorithm, available as both a desktop application and a command-line tool (CLI). Additionally, it is accessible online, providing highly accurate and rapid results comparable to the sensitivity of the gold standard BLAST tool, with the capability of executing multiple tasks simultaneously at speeds of up to 360 times faster [38]. All sequences generated in this study were deposited in GenBank with the accession numbers: MmGmV_06 PP249602; MmGmV_01 PP249603; MmGmV_03 PP249604; MmGmV_02 PP249605; MmGmV_07 PP249606; MmGmV_09 PP249607; and MmGmV_05 PP249608.

### 2.6. Genome Annotation

Comparison of DNA virus sequences in studies were obtained from liver samples (pool-B) of *M. molossus* species and were submitted to an online database DIAMOND and also on the desktop to align the sequences in different taxonomic levels and then obtain the hierarchical structure of the viruses [38].

Furthermore, the comparative analysis of the sequences of the predicted genes was carried out via the online BLASTx program, as it is a protein aligner using a translated DNA sequence. Based on the best results, the sequences were selected to be subsequently aligned [39,40]. All sequences in the analysis were compared with the *Genomoviridae* family from GenBank and subsequently, complete or almost complete genome sequences were selected and aligned using the MAFFT software, v7 [41]. Conserved RCR motifs were predicted using the online software InterProScan (https://www.ebi.ac.uk/interpro/search/sequence/, accessed on 29 June 2023) and and Motif Finder (https://www.genome.jp/tools/motif/, accessed on 25 June 2023), respectively.

### 2.7. Genetic Distance

Genetic distance and its standard error were calculated using the maximum likelihood plus gamma correction and bootstrap model with 1000 replications. MEGA software (Version X) was implemented to estimate distance and genetic diversity [42], as well as SDT v1.2 [43] to calculate identity through a combination of pairwise calculations and perform identity score distribution on pairs of sequences. The initial realignment of sequences involved penalizing gaps—a process carried out with the MUSCLE algorithm [44]. Following the calculation of identity scores for each pair of sequences (pairwise scores), the NEIGHBOR component of PHYLIP was employed to construct a tree. This rooted neighbor-joining phylogenetic tree organizes all sequences based on their likely degrees of evolutionary relatedness.

### 2.8. Phylogenetic Analysis

Phylogenetic trees were constructed using the maximum likelihood approach, using the best evolutionary model (LG + F + R6) inferred with Iq-TREE. Branch support was estimated using a bootstrap test with 1000 replications [45]. The trees were visualized and edited using the Figtree 1.4.2 program (http://tree.bio.ed.ac.uk/software/figtree/).

## 3. Results

### 3.1. Identification of Genomoviridae

We identified seven sequences of genomiviruses (MmGmV-01-PA, MmGmV-02-PA, MmGmV-03-PA, MmGmV-05-PA, MmGmV-06-PA, MmGmV-07-PA, and MmGmV-09-PA) in the pool of liver samples from an *M. molossus* bat. The results of this search are summarized in Table 1. In general, the genomes showed the greatest identity with the CRESS-DNA viral genomes of the *Genomaviridae* family, with BLASTn identity ranging from 74.03 to 98.71%, and BLASTp, from 56.90 to 98.00%.

### 3.2. Genome Characterization

All sequences exhibit two putative open reading frames (ORFs), responsible for encoding the replication-associated protein (Rep) with a size range of 194 to 266 amino acids, and the capsid protein (Cap) spanning 130 to 329 amino acids, as detailed in Table 2. It is important to highlight that, between the 5’ ends of the two ORFs, there is an intergenic region, called large, and in some viruses, between the 3’ ends there is a second intergenic region, called small. The presence of one or two intergenic regions is used to distinguish the type (I and II) Genomovirus genomes. The type I genome contains two intergenic regions, and the type II genome has only one intergenic region (Figure 1). When there is no presence of the small intergenic region, there is a juxtaposition of the ORFs at the 3’ terminals (type II genome), in which there is an intron in the Rep coding region. Possibly the introns within the Rep ORF undergo a splicing process constituting the functional Rep protein, as an example of the genomes of the viral sequences of MmGmV_01-PA, MmGmV-06-PA, MmGmV-07-PA (Figure 1), like Genomovirus and Geminivirus [16,46,47]. The MmGmV-07-PA genome is larger than all viral sequences and contains the essential ORFs, in addition to a putative V2 protein like the *Hypericum japonicum-associated circular DNA virus* (HJasCV) [20].

We also predicted the origin of replication for the seven genomes. The conserved mononucleotide sequence ‘TATATGTGG’ was identified in the MmGmV-06 and MmGmV-07 viral genomes. Conversely, the mononucleotide sequence ‘TATAT’ exhibits conservation only in positions 1 to 5, displaying variability at its ends. This pattern is consistent among the viruses MmGmV-01-PA, MmGmV-02-PA, MmGmV-03-PA, MmGmV-05-PA, and MmGmV-09-PA. Importantly, these distinctive characteristics align with common features observed in other genera within the *Genomoviridae* family, as depicted in Figure 1.

The Rep protein of MmGmV-01 exhibits 60.11% identity with the Rep protein of a virus isolated from US wastewater metagenomic samples (QJB18714.1), while the Cap protein shows 43.61% identity with the Red panda feces-associated *gemycircularvirus* (UBJ26138.1). MmGmV-02-PA and MmGmV-03-PA share 71.11% and 99.48% identity, respectively, with the Rep protein of the *Ouratea duparquetiana* associated *gemykibivirus* (QNI80852.1) and *canine feces-associated gemycircularvirus* (YP_010784661.1). In contrast, the Cap protein exhibits 46.72% and 87.75% identity with metagenomic wastewater (QJB18679.1) and *Red panda feces-associated gemycircularvirus* (UBJ26188.1). The Rep proteins of MmGmV-05-PA and MmGmV-07-PA share 87.31% and 88.61% identity with the Rep proteins of viruses (QTE03605.1 and QTZ83241.1) isolated *Emberiza spodocephala Genomoviridae* sp. Additionally, the Cap protein of MmGmV-05 shares 59.63% identity with the putative coat protein of Dragonfly-associated circular virus 3 (YP_009021851.1), while MmGmV-07-PA shares 82.90% identity with the Cap protein *Genomoviridae* sp. (QXN75602.1).

The Rep protein of MmGmV06-PA is highly similar, with 98% identity, to the RepA *of Pteropus-associated Gemycircularvirus5* isolated from fecal samples of *Pteropus tonganus* bats. Meanwhile, the Cap protein shares 68.25% similarity with the metagenomic animal isolated from haddock tissue (YP_010798116.1). MmGmV09-PA shares 88.50% identity with the Rep protein of the *Emberiza spodocephala* isolate, and its Cap protein shares 58.97% identity with *Genomoviridae* sp. (QXN75548.1).

### 3.3. Analysis of Conserved Motifs

We highlight that the viral genomes of MmGmV-PA belong to circular ssDNA viruses encoding the Rep protein (CRESS-DNA). The Rep protein features crucial motifs for rolling circle replication (RCR), a DNA replication process observed in various single-stranded DNA (ssDNA) viruses [48]. The replication initiator protein (Rep) of geminiviruses is a replicon-specific initiator enzyme and is an essential component of the replisome that carries out viral genome replication its completeness [49].

To gain deeper insights into the sequences, we analyzed conserved motifs within the Rep protein (Figure 2). In the Rep of MmGmV-05-PA, three conserved domains were identified: Geminivirus Rep catalytic domain (Gemini_AL1), putative viral replication protein (Viral_Rep), and Geminivirus Rep protein central domain (Gemini_AL1_M). The Rep of MmGmV-01-PA features Gemini_AL1 and Viral_Rep, while the Rep of MmGmV-02-PA, MmGmV-03-PA, MmGmV-06-PA, and MmGmV-09-PA contains Gemini_AL1 and Gemini_AL1_M. In contrast, the Rep of MmGmV-07-PA exclusively possesses Gemini_AL1. To delve further into the distinctions among the conserved sequences of the Gemini_AL1 geminivirus Rep protein, we scrutinized the motifs present in it. Across all seven MmGmV-PA sequences, the Gemini_AL1 geminivirus Rep catalytic domain, responsible for coding, was consistently identified. The Rep featuring Gemini_AL1 exhibits four conserved regions, denoted as motifs: I, II, GRS domain, and motif III.

In the RCR motif I (six amino acids)—(LLTYxQ), present in the MmGmV Rep, the presence of residue substitutions (“x” for A) in MmGmV (-01, -05, -06 and -09) is identified—(“x” for P) in MmGmV (02 and 07) and MmGmV-03 (“x” for S) in position (5), but we also observed a bulky hydrophobic residue “F” in the sequence of GmV-09 (LFTYAQ), located in second position. RCR motif I is possibly involved in the recognition of iteron sequences associated with the origin of replication and its sequence, and is represented by (uuTYxQ). The letter (u) in the previous representation indicates bulky hydrophobic residues, such as I, L, V, M, F, Y, and W; and the letter (x), any residue [28,30].

Furthermore, the RCR II motif (five amino acids), of MmGmV, which consists of the consensus sequence (xHxHx), presents significant variations within itself and in several species in positions (1, 3 and 5). Histidine is involved in the initial function of coordinating divalent metal ions, Mg^2+^ or Mn^2+^, which are important cofactors for endonuclease activity at the origin of replication [28,30].

We observed the RCR motif III (four amino acids—YATK) conserved for all MmGmV sequences, with a lysine residue that is proposed to measure binding and positioning during catalysis. It is worth noting that motif III (YxxK) has only one catalytic tyrosine and may be involved in the cleavage of dsDNA and the covalent attachment of Rep to the catalytic tyrosine residue at the end of the cleaved product [28,30,31,50].

Rep analysis of MmGmV revealed the GRS IV motif (17 amino acids). GRS IV motifs play a role uniquely in geminiviruses (the conserved sequence is found at the N-terminus of the Rep protein) and in genomoviruses (highlights the importance for prevalence and diversity of genomoviruses in nature) [28,31,47]. The Rep contains the motifs RCR I, II, GRS, and III, which are present in several species of the *Genomoviridae* family, for example three isolates identified in blood samples from patients and blood taken from healthy cattle (HCBI8.215, MSSI2. 225 and HCBI9.212) [13]—the viruses *Hypericum japonicum-associated circular DNA virus* (HJasCV) [20], the *Bemisia-associated genomovirus AdO*, [51], the *Tadarida brasiliensis gemykibivirus* 1 (TbGkyV1), [15], the *Drogonfly* virus (DfasCV-1, -2, -3) [18], and *M. molossus* associated *Gemykibivirus* 1–6 (MAVGs12, 16, 17, 18, 21, 22 and 24) [10].

### 3.4. Genetic Distances and Phylogenetic Inferences

To infer the phylogenetic tree, we used the complete genome and the Rep protein detected in the MmGmV sequences with other references from the *Genomaviridae* family (*Gemycircularvirus, Gemyduguivirus, Gemygorvirus, Gemykibivirus, Gemykolovrus, Gemykrogvirus, Gemykroznavirus, Gemytondvirus, Gemytripvirus*, and *Gemyvongvirus*) (Figure 3). The tree indicated that the sequences grouped into two clades corresponding to the genera *Gemykibivirus* (GmV-01, GmV-02, GmV-03, GmV-05 and GmV-09) and *Gemycirculavirus* (GmV-06 and GmV-07). MmGmV-01 clusters with plant *Gemykibivirus* species 2 (MK947376/QFR58261), sharing 60.89% pairwise identity based on nucleotide sequence with each other. MmGmV_02 showed the greatest relationship with *Gemykibivirus hydro*1 (MK483076/QCS35889) and *Gemykibivirus draga*1 (NC_023872/YP_009021862), sharing 71.41% and 67.91% identity, respectively.

MmGmV_03 grouped with the species of *Gemykibivirus canfam*1 (NC_075339/YP_010784661) isolated in Brazil in 2016. The sequences have 97.70% identity with each other. MmGmV_05-PA grouped with *Emberiza spodocephala Genomoviridae* sp. discovered in birds in China in 2018 (MW182919/QTE03605) exhibiting 84.12% identity. MmGmV_06-PA has 99% identity with the viruses YP_009506608 (NC_038488/YP_009506608) identified in the guano of Flying foxes (*Pteropus tonganus*). MmGmV_07-PA clustered in a clade close to an unclassified sequence named *Genomoviridae*_sp (MW678943/QXN75601) and *Gemycircularvirus-ptero*3 (NC_038486/YP_009506601) with identity of 76.51% and 64.54%. Meanwhile, MmGmV_09-PA showed clustering close to the sequences MW182919 and MmGmV_05-PA, exhibiting identities of 70.99% and 83.18%, respectively. Because, based on the parameter for classifying the *Genomaviridae* family < 78% identity for a new species throughout the genome, as well as for Rep to define the genus, in addition to Rep being established based on phylogenetic analysis [16]. MmGmV_01, 02, and 07 are likely to represent new species within the genus *Gemykibivirus*.

## 4. Discussion

In this comprehensive investigation, our focus was on the exploration of viral presence within liver samples extracted from molossus bats in the municipality of Santarém, situated in the northern region of Brazil’s state of Pará. The study revealed a noteworthy identification of seven distinct sequences termed MmGmV-PA. Through meticulous sequence analyses, these sequences exhibited a significant level of identity ranging from 74.03% to 98.71%. The genomic organization observed shared commonalities with other members of the *Genomoviridae* family, encompassing motifs, conserved nonanucleotides in the ori, and two crucial ORFs responsible for encoding Rep and Cap.

The *Genomoviridae* family, established in 2016, originated with the discovery of the *Sclerotinia sclerotiorum* hypovirulence-associated DNA virus 1 (SsHADV-1) from environmental samples [27]. Since its inception, numerous viruses within this family have been documented across diverse hosts and sample types. Notably, bats of the *Tadarida brasiliensis* and *Molossus molossus* species have been studied extensively, with viral species like *Tadarida brasiliensis gemykibivirus* 1 (TbGkyV1) emerging as the first genomovirus viral sequences from molossus bats in South America [10,15,52,53].

A phylogenetic analysis of the Rep sequences unveiled a classification into two genera: *Gemykibivirus* (MmGmV-01, MmGmV-02, MmGmV-03, MmGmV-05, and MmGmV-09) and *Gemycirculavirus* (MmGmV-06 and MmGmV-07). Utilizing the defined criteria for species demarcation (<78% genome-wide pairwise identity), the MmGmV sequences were found to be associated with four known species (MmGmV-03, MmGmV-05, MmGmV-06, and MmGmV-09) and three novel species (MmGmV-01, MmGmV-02, and MmGmV-07).

Notably, the study recognizes the challenges in comparing results from different metagenomics studies due to various factors such as sample types, methodologies, and bioinformatic analyses. Consequently, comprehensive assessments have been made of pathogenic characteristics and epidemiological data related to *Genomoviridae* in the human sample [12] and a group of CRESS viruses that infect protozoa (*Entamoeba* and *Giardia*) named in the eukaryotic family *Naryaviridae, Nenyaviridae* and *Vilyaviridae*. They are responsible for many cases of human infectious diseases annually in the world [21,22,54]. Further CRESS-DNA viruses remain elusive [26].

Building upon a prior study’s suggestion of a potential association between CRESS DNA viruses in bats and dietary habits, the current findings underscore the importance of monitoring viruses in bats. Bats, being reservoirs for a diverse array of both known and novel viral species, present a crucial focal point for understanding disease dynamics and cross-species transmission [55,56]. In conclusion, these data significantly contribute to an enhanced comprehension of the diversity of CRESS-DNA viruses within molossus bats, particularly expanding our knowledge with the identification of new species within the *Gemykibivirus* genus.

## 5. Conclusions

In conclusion, the investigation contributes to a better understanding of the diversity of CRESS-DNA viruses in bats (*M. molossus*) and expanded knowledge with the discovery of new species within the genus *Gemykibivirus*.

## Figures and Tables

**Figure 1 microorganisms-12-00688-f001:**
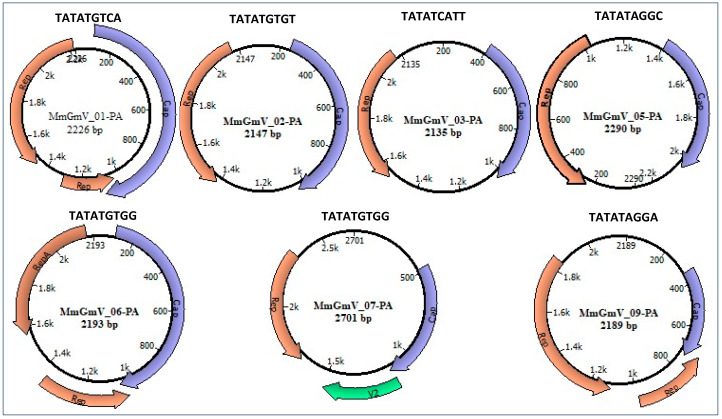
Genome map of the MmGmV-PA. Circular diagrams depicts the genomes of genomoviruses discovered in *Molossus* bats. Distinctively colored regions highlight the identified ORFs in MGmV-PA. The nucleotide sequence positioned above each circle signifies the origin of replication for each virus. Within each circle, the virus name and its genome size in base pairs are presented.

**Figure 2 microorganisms-12-00688-f002:**
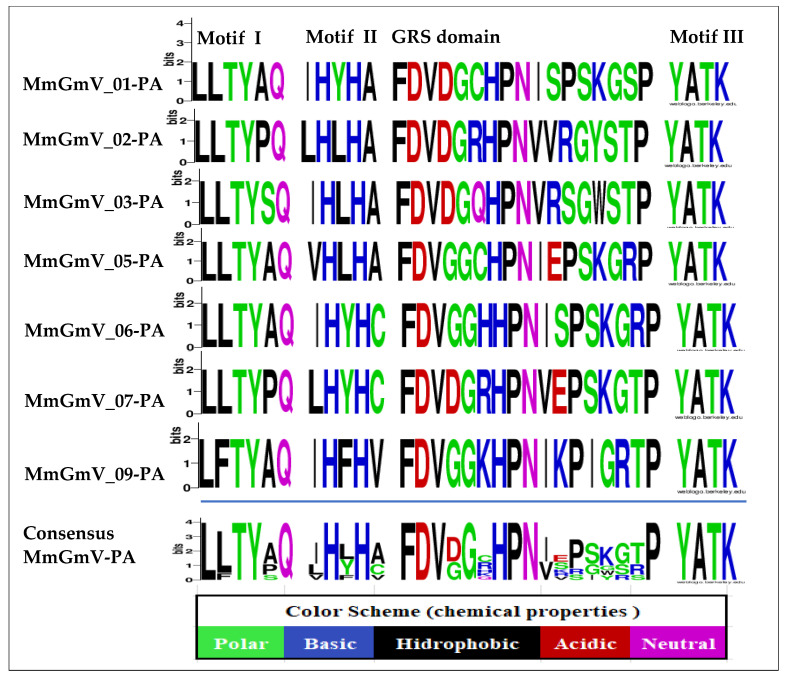
Amino acid signatures of Rep of MmGmV-PA. Colors represent chemical properties of amino acids. The size of each letter in the consensus sequence represents the frequency of amino acids in the sequences of MmGmV-PA.

**Figure 3 microorganisms-12-00688-f003:**
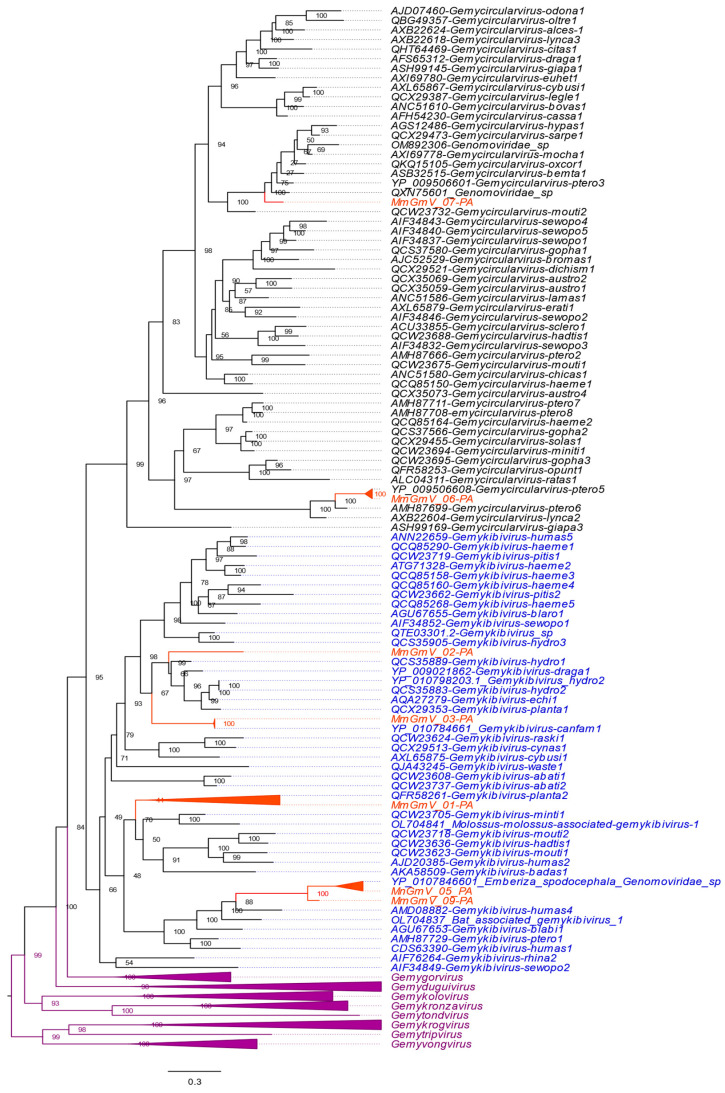
The phylogenetic tree displays MmGmV-PA Rep protein sequences. Species names in black letters are within the genus *Gemycircularvirus*, while the genus *Gemykibivirus* is highlighted in blue letters. The orange triangles represent the grouping of MmGmV-PA viral sequences The remaining families within the *Genomoviridae* family are represented by a collapsed lilac triangle. The numbers on the branches indicate the support inferred by the bootstrap test with 1000 replications.

**Table 1 microorganisms-12-00688-t001:** Genome identity of genomoviruses identified in liver tissue of the *M. molossus*.

ContigID (Genome Size in Base Pairs)	BLAST Search Results						
	Genbank Best-Hit	Coverage		E-Values		Identity	
		BLASTN	BLASTX	BLASTN	BLASTX	BLASTN	BLASTX
Genomovirus01 (2226)	MT205711.1	11%	26%	1 × 10^−15^	2 × 10^−64^	74.03%	56.90%
Genomovirus02 (2147)	NC_076359.1	45%	45%	1 × 10^−125^	3 × 10^−114^	81.05%	63.75%
Genomovirus03 (2135)	NC_075339.1	99%	51%	0.0	0.0	98.71%	86.70%
Genomovirus05 (2290)	MW182919.1	41%	45%	0.0	2 × 10^−148^	88.89%	66.65%
Genomovirus06 (2193)	NC_038488.1	58%	27%	0.0	2 × 10^−135^	93.70%	98.00%
Genomovirus07 (2701)	MW678943.1	43%	44%	0.0	0.0	87.01%	69.85%
Genomovirus09 (2189)	NC_038497.1	42%	26%	0.0	3 × 10^−98^	88.50%	78.65%

**Table 2 microorganisms-12-00688-t002:** Identity score of Rep and Cap of MmGmV.

Sample	Size (Base Pairs)		Best-Hit GenbankID		Coverage		E-Values		Identity	
	Rep	Cap	Rep	Cap	Rep	Cap	Rep	Cap	Rep	Cap
Genomovirus01	257	329	QJB18714.1	UBJ26138.1	73%	91%	4 × 10^−74^	1 × 10^−71^	60.11%	43.61%
Genomovirus02	235	295	QNI80852.1	QJB18679.1	91%	87%	1 × 10^−111^	7 × 10^−77^	71.11%	46.72%
Genomovirus03	194	267	YP_010784661.1	UBJ26188.1	100%	100%	1 × 10^−139^	0.0	99.48%	87.75%
Genomovirus05	266	292	QTE03605.1	YP_009021851.1	74%	74%	5 × 10^−124^	2 × 10^−87^	87.31%	59.63%
Genomovirus06	200	311	YP_009506608.1	YP_010798116.1	100%	100%	2 × 10^−144^	3 × 10^−133^	98%	68.25%
Genomovirus07	215	310	QTZ83241.1	QXN75602.1	93%	100%	1 × 10^−128^	0.0	88.61%	82.90%
Genomovirus09	248	130	QTE03605.1	QXN75548.1	79%	90%	1 × 10^−108^	2 × 10^−44^	79.29%	58.97%

## Data Availability

Data are contained within the article.

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
