# Peer review of "Genomoviruses in Liver Samples of Molossus molossus Bats"

_microorganisms, 2024, doi:10.3390/microorganisms12040688_

Round 1

Reviewer 1 Report

Comments and Suggestions for Authors

Review of “Genomoviruses in liver samples of Molossus molossus bats” (microorganisms-2909365) by da Silva Couto et al.

This paper contributes to describing the diversity of CRESS DNA viruses in the liver of Molossus molossus bats and expanded our knowledge of CRESS DNA viruses with the discovery of new species within the genus Gemykibivirus. The paper is well written and is worth publishing, but I have listed minor fauls/comments/suggestions below.

Line 82: SsHADV-a is mentioned for the first time on line 82, but a description of what this acronym stands for is only given on line 91 (Sclerotinia gemycirculavirus 1, linked to hypovirulence 1). For the sake of correction it should be the other way around.

Line 92: Italicize “Gemykibivirus”.

Lines 115-117: Why did the authors specifically focus on the liver tissue, in particular?

Line 119: Replace “Vortex” by “vortex”.

Line 156: Italicize “M. molossus”.

Line 157: The DIAMOND acronym is only defined on line 157, but it is first used on line 148. It should be the other way around. Furthermore, the text between lines 156 and 157 is awkward, suggesting that DIAMOND is an online database (when it is not). Please correct.

Line 159: Replace “As well as” by “Furthermore,”.

Line 160-161: Replace “DNA alignment” with “translated DNA sequence”. Furthermore, I would delete the whole sentence on line 161 (Therefore, the high similarity between query sequences is described by an alignment) as it complicates the text unnecessarily.

Lines 164-165: Modify the original text (subsequently the sequences were selected and aligned using the MAFFT software, v7, complete or almost complete genomes were aligned) to: “subsequently, complete or almost complete genome sequences were selected and aligned using the MAFFT software, v7”.

Lines 171-180: This section is confusing, and the references given for MegaX, the STD program and MUSCLE are NOT correct. I suggest a thorough checking of the references used. Furthermore, the outcome mentioned (The outcomes are illustrated in a graphical interface through a frequency distribution of paired identities, forming an identity matrix) is never shown…and it should.

Line 192: BLASTp/BLASTn are NOT sequence databases as the sentence suggests. What the authors seem to have done, was to look for homologous protein/nucleotide sequences using the BLASTxand/BLASTn search algorithms, most probably using standard protein and nucleotide databases accessible through the NCBI portal (most probably, as the authors mention GenBank).

Table 1: Authors use “BLAST” in the text, but “Blast” in table one. Please correct to BLAST everywhere.

Line 205: What do the “additional” Orfs mentioned code for, especially since they do not seem to be same in all viral genomes mentioned? I do not see ANY reference to them anywhere else in the text. However, in Fig1 the authors have tentatively named them. A more complete analysis of the products encoded by these Orfs should be included in the text.

Line 232: Italicize “Molossus”.

Lines 240-243: I would strongly advise the authors to change the text in this section as it is confusing. The text should clearly indicate the sequences in question are of viral origin. As it is written, it seems the percentages given are with the Rep protein of a plant and canine feces, which is not correct. The same mistake is again made in lines 254-255. The Cap protein is not homologous to airborne particulate matter, but rather with viral sequences associated with it.

Line 251: Italicize Pteropus (in “Pteropus-associated”).

Line 258-259: The acronym CRESS is mentioned from the Introduction onwards, but it is only defined in line 258-259.

Line 261-262: The sentence “This process is vital for both initiation (replicon) and completion (replisome) of the replication [39]”. In my opinion the text mixes “the process of DNA replication” with the involvement of “the Rep protein” in the initiation of replication and then as a member of the viral replisome. In my opinion it would be simpler to change the text (using the same reference) but simply to mention that the replication initiator protein (Rep) of geminiviruses is a replicon-specific initiator enzyme and is an essential component of the replisome that carries out viral genome replication its completeness.

Line 265: correct “Putative” to “putative” and in line 266 “rep” should be corrected to Rep.

Line 326/334 (Correct its spelling): Italicize Genomoviridae. There is another such fault in line 355, so I suggest a thorough verification of the document should be carried out.

Lines 327-330: The English in this section is awkward. Please correct…eg.: “the viruses (HJasCV) [12]”…and M. molossus (MAVF) [2].”

Lines 341 and 342: The correct designations for the viruses in question are Gemykibivirus hydro1 and Gemykibivirus draga1. In line 344 the authors should have used Gemykibivirus canfam1. I advise towards using https://ictv.global/taxonomy to make sure all the correct viral designations are used. I found some other mistaken designations in the text that I will signal no further.

Line 347: Correctr “Exhibiting” to “exhibiting”.

Line 376: Italicize Sclerotinia sclerotiorum. Italicize Tadarida brasiliensis in line 380.

Line 381: I would advise against using the word “isolated” because, for me, it suggests the virus have been isolated under lab conditions…which is not true. We are talking about viral sequences…not viral isolates.

Comments on the Quality of English Language

I appended comments in my reply to the authors. they included only minor suggestions 

Author Response

Dear Editor

Microorganisms

We are glad to know that our manuscript, “Genomoviruses in liver samples of Molossus molossus bats,” has been considered within the scope of this journal.

We deeply appreciated the detailed and insightful comments made by the referees. We would also like to thank the editorial board for processing our manuscript. All efforts were made to enhance the quality of the manuscript and to allow for its expedient publication. We are sure that this paper has greatly improved after making the modifications suggested by the reviewers.

Sincerely

Elcio Leal

Reviewer 1

1) Line 82: SsHADV-a is mentioned for the first time on line 82, but a description of what this acronym stands for is only given on line 91 (Sclerotinia gemycirculavirus 1, linked to hypovirulence 1). For the sake of correction it should be the other way around.

RESP: The acronym SsHADV-a stands for Sclerotinia gemycirculavirus1. We changed this in the new version of the manuscript.

2) Line 92: Italicize “Gemykibivirus”.

RESP: Italics have already been applied to the word “Gemykibivirus”.

3) Lines 115-117: Why did the authors specifically focus on the liver tissue, in particular?

RESP: Many metagenomic studies conducted using swabs or fecal samples are prone to contamination from mucosal or intestinal flora. For this reason, we opted to use liver samples. However, in the case of the genomovirus we discovered, the likely host appears to be a parasitic organism present in the liver.

4) Line 119: Replace “Vortex” by “vortex”.

RESP: done.

5) Line 156: Italicize “M. molossus”.

RESP: We changed M. molossus” to italic “M. molossus”.

6) Line 157: The DIAMOND acronym is only defined on line 157, but it is first used on line 148. It should be the other way around. Furthermore, the text between lines 156 and 157 is awkward, suggesting that DIAMOND is an online database (when it is not). Please correct.

RESP: We changed this in the new version of the masnucript.

7) Line 159: Replace “As well as” by “Furthermore,”.

RESP: The word “As well” was replaced with “Furthermore,”.

8) Line 160-161: Replace “DNA alignment” with “translated DNA sequence”. Furthermore, I would delete the whole sentence on line 161 (Therefore, the high similarity between query sequences is described by an alignment) as it complicates the text unnecessarily.

RESP: The phrase “DNA alignment” was replaced with “translated DNA sequence”. The phrase “Therefore, the high similarity between query sequences is described by an alignment” was also removed from the text.

9) Lines 164-165: Modify the original text (subsequently the sequences were selected and aligned using the MAFFT software, v7, complete or almost complete genomes were aligned) to: “subsequently, complete or almost complete genome sequences were selected and aligned using the MAFFT software, v7”.

RESP: The original text was modified to “subsequently, complete or almost complete genome sequences were selected and aligned using the MAFFT software, v7”

10) Lines 171-180: This section is confusing, and the references given for MegaX, the STD program and MUSCLE are NOT correct. I suggest a thorough checking of the references used. Furthermore, the outcome mentioned (The outcomes are illustrated in a graphical interface through a frequency distribution of paired identities, forming an identity matrix) is never shown…and it should.

 RESP: We have updated the references. The sentence: The outcome are illustrated … refers to the similarity plot generated by STD software. This sentence was removed from the new version of the manuscript.

12) Line 205: What do the “additional” Orfs mentioned code for, especially since they do not seem to be same in all viral genomes mentioned? I do not see ANY reference to them anywhere else in the text. However, in Fig1 the authors have tentatively named them. A more complete analysis of the products encoded by these Orfs should be included in the text.

RESP: We have tentatively identified two types of viral genomes in the genomevirus sequences, designated as Type I and Type II, which are distinguished by overlapping ORFs. For instance:

It is important to note that between the 5' ends of the two ORFs, there exists an intergenic region, referred to as large, and in some viruses, between the 3' ends, there is a second intergenic region, referred to as small. The presence of one or two intergenic regions is utilized to differentiate between Type I and Type II Genomovirus genomes. The Type I genome contains two intergenic regions, while the Type II genome possesses only one intergenic region (refer to Figure 1). In instances where the small intergenic region is absent, the ORFs at the 3' terminals are juxtaposed (Type II genome), resulting in an intron within the Rep coding region. It is possible that the introns within the Rep ORF undergo a splicing process, forming the functional Rep protein. This phenomenon is exemplified in the genomes of viral sequences such as MmGmV_01-PA, MmGmV-06-PA, and MmGmV-07-PA (refer to Figure 1), similar to Genomoviruses and Geminiviruses. Notably, the MmGmV-07-PA genome is larger than all other viral sequences and contains essential ORFs, along with a putative V2 protein akin to the Hypericum japonicum-associated circular DNA virus (HJasCV).

13) Line 232: Italicize “Molossus”.

RESP: We changed “Molossus” to an italic "Molossus.”.

14) Lines 240-243: I would strongly advise the authors to change the text in this section as it is confusing. The text should clearly indicate the sequences in question are of viral origin. As it is written, it seems the percentages given are with the Rep protein of a plant and canine feces, which is not correct. The same mistake is again made in lines 254-255. The Cap protein is not homologous to airborne particulate matter, but rather with viral sequences associated with it.

RESP: We changed the text to clarify.

15) Line 251: Italicize Pteropus (in “Pteropus-associated”).

RESP: We changed it.

16) Line 258-259: The acronym CRESS is mentioned from the Introduction onwards, but it is only defined in line 258-259.

RESP: Circular Rep-encoding Single-Stranded DNA (CRESS) is now in the introduction.

17) Line 261-262: The sentence “This process is vital for both initiation (replicon) and completion (replisome) of the replication [39]”. In my opinion the text mixes “the process of DNA replication” with the involvement of “the Rep protein” in the initiation of replication and then as a member of the viral replisome. In my opinion it would be simpler to change the text (using the same reference) but simply to mention that the replication initiator protein (Rep) of geminiviruses is a replicon-specific initiator enzyme and is an essential component of the replisome that carries out viral genome replication its completeness.

RESP: The sentence in the text was replaced by: "The replication initiator protein (Rep) of geminiviruses is a replicon-specific initiator enzyme and is an essential component of the replisome that carries out viral genome replication.”.

18) Line 265: correct “Putative” to “putative” and in line 266 “rep” should be corrected to Rep.

RESP: The word “Putative” written with the capital letter “P” was replaced by the correct writing “putative” in lowercase “p”.

19) Line 326/334 (Correct its spelling): Italicize Genomoviridae. There is another such fault in line 355, so I suggest a thorough verification of the document should be carried out.

RESP: The cursive spellings “Genomoviridae” were graded in italics “Genomoviridae”.

20) Lines 327-330: The English in this section is awkward. Please correct…eg.: “the viruses (HJasCV) [12]”…and M. molossus (MAVF) [2].”

RESP: “Hypericum japonicum-associated circular DNA virus (HJasCV) [20], and M. molossus associated Gemykibivirus 1-6 (MAVGs12, 16, 17, 18, 21, 22 and 24) [10]”.

21) Lines 341 and 342: The correct designations for the viruses in question are Gemykibivirus hydro1 and Gemykibivirus draga1. In line 344 the authors should have used Gemykibivirus canfam1. I advise towards using https://ictv.global/taxonomy to make sure all the correct viral designations are used. I found some other mistaken designations in the text that I will signal no further.

RESP: The correct spelling of viruses has been replaced according to ICTV: “Gemykibivirus canfam1”, and “Gemykibivirus draga1”.

22) Line 347: Correctr “Exhibiting” to “exhibiting”.

RESP: This was changed in the new version of the manuscript.

23) Line 376: Italicize Sclerotinia sclerotiorum. Italicize Tadarida brasiliensis in line 380.

RESP: The cursive spellings “Sclerotinia sclerotiorum; Tadarida brasiliensis” were graded in italics “Sclerotinia sclerotiorum; Tadarida brasiliensis”.

23)Line 381: I would advise against using the word “isolated” because, for me, it suggests the virus have been isolated under lab conditions…which is not true. We are talking about viral sequences…not viral isolates.

RESP: The word “isolated” was replaced by “viral sequences”

Reviewer 2 Report

Comments and Suggestions for Authors

The article entitled "Genomoviruses in liver samples of Molossus molossus bats", presents a voluble study on the diversity of CRESS DNA viruses in Velvety free-tailed bats and expanded knowledge with the discovery of new species within the genus Gemykibivirus.

Minor comments:

Keywords section - Molossus molossus is also in the title and must be deleted from the keywords. It can be replaced by Velvety free-tailed bat.

Line 101-102, please add the reference after ''based on external characteristics''.

Author Response

Dear Editor

Microorganisms

We are glad to know that our manuscript, “Genomoviruses in liver samples of Molossus molossus bats,” has been considered within the scope of this journal.

We deeply appreciated the detailed and insightful comments made by the referees. We would also like to thank the editorial board for processing our manuscript. All efforts were made to enhance the quality of the manuscript and to allow for its expedient publication. We are sure that this paper has greatly improved after making the modifications suggested by the reviewers.

Sincerely

Elcio Leal

REVIEWER 2:

The article entitled "Genomoviruses in liver samples of Molossus molossus bats", presents a voluble study on the diversity of CRESS DNA viruses in Velvety free-tailed bats and expanded knowledge with the discovery of new species within the genus Gemykibivirus.

Minor comments:

Keywords section - Molossus molossus is also in the title and must be deleted from the keywords. It can be replaced by Velvety free-tailed bat.

RESP: We replaced by “Velvety free-tailed bat”.

Line 101-102, please add the reference after ''based on external characteristics''.

RESP: The text was substantially changed in this section. We also added some new references.